# Feedback Control of Quantum Correlations in a Cavity Magnomechanical System with Magnon Squeezing

**DOI:** 10.3390/e25101462

**Published:** 2023-10-18

**Authors:** Mohamed Amazioug, Shailendra Singh, Berihu Teklu, Muhammad Asjad

**Affiliations:** 1LPTHE-Department of Physics, Faculty of Sciences, Ibnou Zohr University, Agadir 80000, Morocco; amazioug@gmail.com; 2Process Systems Engineering Centre (PROSPECT), Research Institute of Sustainable Environment (RISE), Universiti Teknologi Malaysia, Johor Bahru 81310, Johor, Malaysia; singhshailendra3@gmail.com; 3Department of Applied Mathematics and Sciences, Khalifa University, Abu Dhabi 127788, United Arab Emirates; muhammad.asjad@ku.ac.ae; 4Center for Cyber-Physical Systems (C2PS), Khalifa University, Abu Dhabi 127788, United Arab Emirates

**Keywords:** cavity magnomechanics, coherent feedback, entanglement, steerability

## Abstract

We suggest a method to improve quantum correlations in cavity magnomechanics, through the use of a coherent feedback loop and magnon squeezing. The entanglement of three bipartition subsystems: photon-phonon, photon-magnon, and phonon-magnon, is significantly improved by the coherent feedback-control method that has been proposed. In addition, we investigate Einstein-Podolsky-Rosen steering under thermal effects in each of the subsystems. We also evaluate the scheme’s performance and sensitivity to magnon squeezing. Furthermore, we study the comparison between entanglement and Gaussian quantum discord in both steady and dynamical states.

## 1. Introduction

Entanglement and Einstein–Podolsky–Rosen (EPR) steering are two quantum resources in the field of quantum information processing and communication. Quantum entanglement plays a crucial role in various quantum information processing tasks, such as quantum teleportation [1], superdense coding [2], telecloning [3] and quantum cryptography [4]. Many schemes have been proposed over the past decades for processing quantum information such as spins [5,6], ions [7,8,9,10], atoms [11,12,13,14,15], photons [16,17,18,19,20,21,22], and phonons [23,24]. Besides, quantum steering is a concept closely related to entanglement and was introduced by Schrödinger in the context of the EPR parado [25,26] and it can be asymmetric (one-way), and symmetric (two-way) [27]. Quantum steering is a form of quantum correlation that lies between the concepts of entanglement and Bell nonlocality and stronger than entanglement [28] but weaker than the violation of Bell’s inequality [29] and can be observed in various quantum systems, including optomechanical systems [30,31]. It has applications in various areas such as quantum key distribution [32,33], where it can be used to verify the security of the communication channel.

In recent years, magnons, which are quanta of collective spin excitations in materials like yttrium iron garnet $\left(Y_{3}{\mathrm{Fe}}_{5}O_{12},\mathrm{YIG}\right)$ [34,35,36,37], have gained significant attention in recent years due to their desirable properties such as high spin density, low damping rate, and tunability. The field of cavity magnomechanics has emerged as a robust platform for studying magnons, where a YIG sphere or similar ferrimagnetic crystal is coupled with a microwave cavity [38,39]. In cavity magnetomechanics, a magnetostrictive force mediates the interaction between a ferromagnet’s (or ferrimagnet’s) vibratory deformation mode and a magnon mode (spin wave). Additionally, it interacts magnetically with a microwave cavity mode to communicate. The magnetostrictive interaction for huge ferromagnetic materials can be compared to radiation pressure as a dispersive interaction. In this case, the mechanical mode’s frequency is substantially lower than the magnon’s frequency [40,41].

In this paper, we consider coherent feedback technique [42,43,44] to enhance the entanglement and steerability in a cavity magnomechanics consisting of a cavity containing YIG sphere with the magnon self-Kerr nonlinearity as illustrated in Figure 1. The use of a feedback model in the cavity magnomechanical system with magnon squeezing allows to actively control and optimize the system’s behavior for specific applications. The nonlinearity in the system plays a central role in enabling the coupling between magnons and phonons, making it possible to manipulate and enhance squeezing effects and has been extensively studied in the field of quantum optics and quantum information science [45,46,47]. As a result, feedback models are valuable tools in the study of quantum effects and quantum technologies, where precise control of quantum states is essential for various applications. The YIG sphere, with its magnon self-Kerr nonlinearity, implies that the nonlinearity arises from the interaction between magnons within the YIG material. We find a significant enhancement of quantum correlations via magnon squeezing which is generated by using the magnon self-Kerr nonlinearity [48,49]. The magnon self-Kerr nonlinearity [50,51,52,53,54] can be achieved by coupling the magnon mode to a superconducting qubit [55]. In order to quantify the quantum entanglement of three bipartitions subsystems, we take into account the logarithmic negativity [56,57]. The steerability of the subsystem *A* by the first subsystem *B* is used to quantify the steerability between two modes. In the presence of the magnon self-Kerr nonlinearity, we explore the strengthening of nonclassical correlations via coherent feedback method. By employing the coherent feedback technique and evaluating the Gaussian quantum discord, we can investigate how the nonclassical correlations are enhanced and their robustness against thermal effects when β=π. This analysis allows us to explore the role of the feedback technique and the impact of magnon self-Kerr nonlinearity on the system’s quantum correlations, both in steady and dynamical states [58].

The paper is organized as follows. In Section 2, we give the explicit expression of the Hamiltonian and the corresponding nonlinear quantum Langevin equations of the system. In Section 3, we provide the linearized quantum Langevin equations for the system. We present a method in Section 4 to quantify entanglement for two-mode continuous-variable (CV) systems, Gaussian quantum steering and Gaussian quantum discord. The results and discussions are given in Section 5. Concluding remarks are given in Section 6.

## 2. Model

We consider a cavity magnomechanics driven by a single coherent laser source and a microwave cavity with coherent feedback as depicted in Figure 1, where a YIG sphere with 250 μm diameter (Ref. [40]) is placed inside the cavity and is used to couple magnons (collective excitations of spins in a magnetic material) with cavity photons. The coupling between magnons and cavity photons in this system occurs through the magnetic dipole interaction. The magnetostrictive interaction mediates the coupling between the magnons and cavity phonons. This coupling leads to the magnon-induced deformation of the YIG sphere’s geometric structure and the formation of vibrational modes, as well as the reverse effect where the vibrational modes of the sphere influence the magnons and cavity photons, and vice versa [60]. It’s worth noting that the influence of radiation pressure is considered to be insignificant in this system due to the small size of the YIG sphere compared to the microwave wavelength. In Figure 1, an asymmetric beam splitter is used to divide an input laser beam with frequency $\omega_{0}$ and amplitude E into two parts. The beam splitter has reflection and transmission coefficients denoted as τ and ψ, respectively. These coefficients are real and satisfy the relation $\psi^{2}+\tau^{2}=1$, indicating that no energy is absorbed within the beam splitter itself. Based on the beam splitter properties, the transmitted portion of the input laser beam has an amplitude of ψE, while the reflected portion has an amplitude of −τE. The transmitted part of the input laser beam, with an amplitude of ψE, is used to pump a cavity. A cavity typically consists of two mirrors facing each other, forming an optical resonator. The cavity’s output field represents the light that escapes the cavity through one of the mirrors. In this setup, a portion of the cavity’s output field is sent back into the cavity using a totally reflecting mirror, denoted as *M*, and the asymmetric beam splitter. The totally reflecting mirror *M* reflects all incident light back with an amplitude equal to its incident amplitude. The beam splitter, being asymmetric, will transmit a portion of the light incident upon it and reflect the remaining part. By sending a portion of the cavity’s output field back into the cavity, the setup can create an optical feedback loop, allowing for additional interactions and manipulations of the laser beam within the cavity, as depicted in Figure 1.

The Hamiltonian of the system has the form (with ℏ=1)
(1)$$\begin{array}{ccc} H & = & \omega_{a}a^{†}a+\omega_{b}b^{†}b+\frac{\omega_{m}}{2}\left(q^{2}+p^{2}\right)+\xi {\left(b^{†}b\right)}^{2}+g_{b}b^{†}bq+g_{a}\left(a+a^{†}\right)\left(b+b^{†}\right)\\ & + & i\Omega \left(b^{†}e^{-i\omega_{0}t}-be^{i\omega_{0}t}\right)+\psi E(a^{†}e^{-i\omega_{0}t}+ae^{i\omega_{0}t}). \end{array}$$

Here we have represented the annihilation (creation) operators of the cavity and magnon modes by *a* ($a^{†}$) and *b* ($b^{†}$) ($[O,O^{†}]\;=\;1$, O=a,b) whereas *q* and *p* ([q,p]=i) stands for the mechanical mode’s dimensionless position and momentum quadratures with $\omega_{a}$, $\omega_{b}$, and $\omega_{m}$ are the resonance frequencies for the cavity, magnon, and mechanical modes, respectively.

Magnon squeezing refers to the phenomenon in which the quantum fluctuations of magnons, which are collective excitations of spins in a magnetic material, are reduced below the standard quantum limit. The self-Kerr term $\xi {\left(b^{†}b\right)}^{2}$, plays a crucial role in producing magnon squeezing. The coefficient ξ represents the self-Kerr coefficient, which determines the strength of the self-interaction of magnons. This nonlinearity leads to an interaction between different magnon modes, which can generate squeezing. The self-Kerr term $\xi {\left(b^{†}b\right)}^{2}$ can induce squeezing by modifying the quantum state of the magnons. In such systems magnon frequency mainly depends upon both the gyromagnetic ratio κ as well as the external bias magnetic field *H* i.e., $\omega_{b}=\kappa H$. Moreover, we can significantly improve the magnomechanical interaction by directly driving the YIG sphere with a microwave source [50,61]). In the strong coupling regime, the decay rates of the cavity and magnon modes denoted as $\gamma_{a}$ and $\gamma_{b}$ are significantly greater than the coupling rate $g_{a}$ between the magnon and microwave, $g_{a}>\gamma_{a},\gamma_{b}$. In the rotating-wave approximation (RWA) of the system at the drive frequency ($\omega_{0}$), we have simplified $g_{a}\left(a+a^{†}\right)\left(b+b^{†}\right)\to g_{a}\left(ab^{†}+a^{†}b\right)$ (valid when $\omega_{a},\omega_{b}\gg g_{a},\gamma_{a},\gamma_{b}$, which is easily satisfied [40]). The parameter $\Omega =\frac{\sqrt{5}}{4}\kappa \;\sqrt{N}B_{0}$ denotes the Rabi frequency as described in [62], where κ/2π=28 GHz/T and the total number of spins N=ρV where *V* is the sphere’s volume and $\rho =4.22\times 10^{27}$ m${}^{-3}$ is the spin density of the YIG. With the approximation of low-lying excitations, $\langle b^{†}b\rangle \ll 2Ns$, where $s=\frac{5}{2}$ denote the spin number of the ground state Fe${}^{3+}$ ion in YIG. In that case, the quantum Langevin equations (QLEs) can be used to describe the system’s entire dynamics in the presence of coherent feedback and noise as
(2)$$\begin{array}{ccc} \dot{a} & = & -\left(i\Delta_{fb}+\gamma_{fb}\right)a-ig_{a}b-i\psi E+{\left(2\gamma_{a}\right)}^{\frac{1}{2}}a_{fb}^{\mathrm{in}},\\ \dot{b} & = & -\left(i\Delta_{b}+\gamma_{b}\right)b-ig_{a}a-ig_{b}bq-2i\xi b^{†}bb+\Omega +\sqrt{2\gamma_{b}}b^{\mathrm{in}},\\ \dot{q} & = & \omega_{m}p,\\ \dot{p} & = & -\omega_{m}q-\gamma_{m}p-g_{b}b^{†}b+\phi , \end{array}$$
where $\Delta_{b}=\omega_{b}-\omega_{0}+\xi$, $\gamma_{m}$ is the mechanical damping rate, $\gamma_{fb}=\gamma_{a}\left(1-2\tau \cos\beta \right)$ is the modified cavity decay rate and $\Delta_{fb}=\Delta_{a}-2\gamma_{a}\tau \sin\beta$ is the effective detuning with $\Delta_{a}=\omega_{a}-\omega_{0}$. The operator $a_{fb}^{in}$ describes the effective input noise operator in the presence of coherent feedback and corresponding description is based on input-output theory [63]. Specifically it can be written as $a_{fb}^{in}=\tau e^{i\beta }a^{out}+\psi a^{in}$, where $a^{in}$ is the input noise operator associated with microwave mode with only non-zero correlations $\langle a^{in†}\left(t\right)a^{in}\left(t^{\prime }\right)\rangle =n_{a}\left(\omega_{a}\right)\delta \left(t-t^{\prime }\right)$ and $\langle a^{in}\left(t\right)a^{in†}\left(t^{\prime }\right)\rangle =\left(n_{a}\left(\omega_{a}\right)+1\right)\delta \left(t-t^{\prime }\right)$. The corresponding correlation functions for the effective input noise operator $a_{fb}^{in}$ for the microwave mode can be written as
(3)$$\begin{array}{ccc} \langle a_{fb}^{\mathrm{in}}\left(t\right)\;a_{fb}^{\mathrm{in}†}\left(t^{\prime }\right)\rangle & = & \psi^{2}{\left|1-\tau e^{i\beta }\right|}^{2}\left[n_{a}\left(\omega_{a}\right)+1\right]\;\delta \left(t-t^{\prime }\right),\\ \langle a_{fb}^{\mathrm{in}†}\left(t\right)\;a_{fb}^{\mathrm{in}}\left(t^{\prime }\right)\rangle & = & \psi^{2}{\left|1-\tau e^{i\beta }\right|}^{2}n_{a}\left(\omega_{a}\right)\;\delta \left(t-t^{\prime }\right). \end{array}$$Moreover, $b^{\mathrm{in}}$ and ϕ represent the noise sources associated with the magnon and mechanical modes, respectively. These noise operators have zero mean and are characterized by specific correlation functions with the following correlation functions [64]
(4)$$\begin{array}{ccc} \langle b^{\mathrm{in}}\left(t\right)\;b^{\mathrm{in}†}\left(t^{\prime }\right)\rangle & = & \left[n_{b}\left(\omega_{b}\right)+1\right]\;\delta \left(t-t^{\prime }\right), \end{array}$$
(5)$$\begin{array}{ccc} \langle b^{\mathrm{in}†}\left(t\right)\;b^{\mathrm{in}}\left(t^{\prime }\right)\rangle & = & n_{b}\left(\omega_{b}\right)\;\delta \left(t-t^{\prime }\right), \end{array}$$
(6)$$\begin{array}{ccc} \langle \phi \left(t\right)\phi \left(t^{\prime }\right)\;+\;\phi \left(t^{\prime }\right)\phi \left(t\right)\rangle /2\;\; & ≃ & \;\;\gamma_{b}\left[2n_{m}\left(\omega_{m}\right)+1\right]\delta \left(t-t^{\prime }\right). \end{array}$$The mechanical quality factor $Q=\omega_{m}/\gamma_{m}\;\;\gg \;1$ is large for a Markovian approximation [65], where $n_{j}\left(\omega_{j}\right)\;=\;{\left[\exp\left(\frac{\hbar \omega_{j}}{k_{B}T}\right)-1\right]}^{-1}$(j=a,b,m) are the equilibrium mean thermal photon, magnon, and phonon number, respectively.

## 3. Linearization of Quantum Langevin Equations

The Heisenberg–Langevin in Equation (Equation 2) are non-linear in nature and generally cannot be solved analytically in most cases. One common approach to solve analytically is to use a linearization scheme, which involves re-writing the mode operators as a sum of the steady state average and the quantum fluctuation operator as O=〈O〉+δO (O=a,b,q,p). This allows us to treat the system perturbatively by neglecting second-order fluctuation terms. For a strongly driven magnon mode (|〈b〉|≫1) and a cavity field with large amplitudes |〈a〉|≫1, the steady-state solutions can be obtained by solving the resulting equations as
(7)$$\begin{array}{ccc} \langle b\rangle & = & \frac{\Omega -ig_{a}\langle a\rangle }{i{\tilde{\Delta }}_{b}+\gamma_{b}}, \end{array}$$
(8)$$\begin{array}{ccc} \langle a\rangle & = & -\frac{ig_{a}\langle b\rangle +i\psi E}{i\Delta_{fb}+\gamma_{fb}} \end{array}$$
and for $|{\tilde{\Delta }}_{b}\left|,\right|\Delta_{fb}|\gg \gamma_{fb},\gamma_{b}$, one gets
(9)$$\langle b\rangle ≃\frac{i\Omega \Delta_{fb}-i\psi E}{g_{a}^{2}-{\tilde{\Delta }}_{b}\Delta_{fb}},$$
where ${\tilde{\Delta }}_{b}=\Delta_{b}+g_{b}\langle q\rangle +2\xi {\left|\langle b\rangle \right|}^{2}$ is the effective magnon-drive detuning including the frequency shift due to the magnomechanical interaction with $\xi \equiv -2i\xi {\langle b\rangle }^{2}$, and $G_{b}=i\sqrt{2}g_{b}\langle b\rangle$ is the effective magnomechanical coupling raFigurete, where $\langle q\rangle =-\frac{g_{b}}{\omega_{m}}{\langle b\rangle }^{2}$. The linearized QLEs describing the quadrature fluctuations $\delta X_{a}=\left(\delta a+\delta a^{†}\right)/\sqrt{2},\;\delta Y_{a}=i\left(\delta a^{†}-\delta a\right)/\sqrt{2},\delta X_{b}=\left(\delta b+\delta b^{†}\right)/\sqrt{2},\delta Y_{b}=i\left(\delta b^{†}-\delta b\right)/\sqrt{2},\delta q$ and δp can be written in compact matrix form as
(10)$$\dot{u}\left(t\right)=Lv\left(t\right)+\mu \left(t\right),$$
with $v\left(t\right)={\left[\delta X_{a}\left(t\right),\delta Y_{a}\left(t\right),\delta X_{b}\left(t\right),\delta Y_{b}\left(t\right),\delta q(t),\delta p(t)\right]}^{T}$ is vector of quadrature fluctuation operators, $\mu \left(t\right)={\left[\;\sqrt{2\gamma_{a}}X_{a}^{\mathrm{in}}\left(t\right),\sqrt{2\gamma_{a}}Y_{a}^{\mathrm{in}}\left(t\right),\sqrt{2\gamma_{b}}X_{b}^{\mathrm{in}}\left(t\right),\sqrt{2\gamma_{b}}Y_{b}^{\mathrm{in}}\left(t\right),0,\phi \left(t\right)\right]}^{T}$ is the vector of input noise operators, and the drift matrix L can be given by
(11)$$L=\left(\begin{array}{cccccc} -\gamma_{fb} & \Delta_{fb} & 0 & g_{a} & 0 & 0\\ -\Delta_{fb} & -\gamma_{fb} & -g_{a} & 0 & 0 & 0\\ 0 & g_{a} & -\gamma_{b}+\xi & {\tilde{\Delta }}_{b} & -G_{b} & 0\\ -g_{a} & 0 & -{\tilde{\Delta }}_{b} & -\gamma_{b}-\xi & 0 & 0\\ 0 & 0 & 0 & 0 & 0 & \omega_{m}\\ 0 & 0 & 0 & G_{b} & -\omega_{m} & -\gamma_{m} \end{array}\right).$$The system reaches its stable and steady-state condition only if the real parts of all eigenvalues of the drift matrix L are negative [66]. The drift matrix in Equation (Equation 11) is provided under the condition where $|{\tilde{\Delta }}_{b}\left|,\right|\Delta_{fb}|\gg \gamma_{fb},\gamma_{b}$. Furthermore, it is mentioned that later it will be shown that $|{\tilde{\Delta }}_{b}\left|,\right|\Delta_{fb}|≃\omega_{m}\gg \gamma_{fb},\gamma_{b}$ [as shown in Figure 1]. These conditions are considered optimal for the presence of all bipartite entanglements in the system. It is important to note that Equation (Equation 7) is intrinsically nonlinear due to the presence of ${\tilde{\Delta }}_{b}$, which itself depends on ${\left|\langle b\rangle \right|}^{2}$. However, for a given value of ${\tilde{\Delta }}_{b}$ (one can straightforwardly achieve the value of ${\tilde{\Delta }}_{b}$) and, consequently, the value of $G_{b}$ by adjusting the bias magnetic field.

## 4. Entanglement, Steerability and Discord

The continuous variable (CV), three-mode Gaussian steady-dynamical state of the system’s quantum fluctuations is entirely described by a 6×6 covariance matrix (CM) V, is written as [67]
(12)$$\dot{V}\left(t\right)=LV\left(t\right)+V\left(t\right)L^{T}+K,$$
with $V_{ij}=\frac{1}{2}\langle v_{i}\left(t\right)v_{j}\left(t^{\prime }\right)+v_{j}\left(t^{\prime }\right)v_{i}\left(t\right)\rangle$ (i,j=1,2,...,6) and $K=\mathrm{diag}\left[\gamma_{a}\psi^{2}|1-\tau e^{i\beta }{|}^{2}\left(2n_{a}+1\right),\gamma_{a}\psi^{2}{\left|1-\tau e^{i\beta }\right|}^{2}\left(2n_{a}+1\right),\gamma_{b}\left(2n_{b}+1\right),\gamma_{b}\left(2n_{b}+1\right),0,\gamma_{m}\left(2n_{m}+1\right)\right]$ is the diffusion matrix, which is defined through $\langle \mu_{i}\left(t\right)\mu_{j}\left(t^{\prime }\right)+\mu_{j}\left(t^{\prime }\right)\mu_{i}\left(t\right)\rangle /2=K_{ij}\delta \left(t-t^{\prime }\right)$. The initial state of the system is considered in the vacuum state. The covariance matrix $\sigma_{AB}$ of two modes *A* and *B* may be written as
(13)$$\sigma_{AB}=\left(\begin{array}{cc} A & C\\ C^{T} & B \end{array}\right).$$The 2×2 sub-matrices A and B in Equation (Equation 13) represent the autocorrelations of the two modes, while the 2×2 sub-matrix C in Equation (Equation 13) defines the cross-correlations of the two modes. Quantifying and characterizing quantum correlations in multipartite quantum systems, such as in cavity magnomechanics, is indeed a challenging task. Various measures have been proposed to quantify entanglement, which is one form of quantum correlation. One commonly used measure for continuous variable (CV) systems is the logarithmic negativity, $E_{N}$ [56,57]
(14)$$E_{N}=\max\left[0,-\log\left(2\Lambda^{-}\right)\right],$$
with $\Lambda^{-}=\sqrt{X-{\left(X^{2}-4\det\sigma_{AB}\right)}^{1/2}}/\sqrt{2}$ being the covariance matrix’s smallest symplectic eigenvalues, which correspond to the partially transposed state of the two modes, with X=detA+detB−2detC. If the logarithmic negativity (EN) is greater than 0, the two subsystems are entangled. On the other hand, if the smallest symplectic eigenvalue ($\Lambda^{-}\geq 1/2$), then the state is separable. The quantum steering quantifier is a further quantum correlation quantifier that is crucial in cavity magnomechanics. The steerability of Bob (B) by Alice (A)
(A→B) for a $\left(n_{A}+n_{B}\right)$ mode Gaussian state can be quantified by [27]
(15)$$S^{A\to B}\left(\sigma_{AB}\right)=\max\left[0,\frac{1}{2}ln\left(\frac{\det A}{4\det\sigma_{AB}}\right)\right],$$The steerability of Alice by Bob $\left[S^{B\to A}\left(\sigma_{AB}\right)\right]$ can be obtained by swapping the roles of *A* and *B*. It is interesting to note that while a steerable state is always a non-separable state, the reverse is not always true. Thus we have two possibilities between *A* and *B*: (*i*) if $S^{A\to B}=S^{B\to A}=0$ Alice can’t steer Bob and vice versa even if they are entangled (i.e., no-way steering), (ii) if $S^{A\to B}>0$ and $S^{B\to A}=0$ or $S^{A\to B}=0$ and $S^{B\to A}>0$ as one-way steering, i.e., Alice can steer Bob but Bob can’t steer Alice and vice versa, and (iii) if $S^{A\to B}=S^{B\to A}>0$ Alice can steer Bob and vice versa (i.e., two-way steering). In addition, the measurement of Gaussian Steering is always bounded by the entanglement. In addition to examining the two mode Gaussian state’s asymmetric steerability, we also present steering asymmetry, given by
(16)$$S\left(AB\right)=|S^{A\to B}-S^{B\to A}|.$$The quantum correlations (nonclassical correlations) beyond entanglement in bipartite system can be measured via the Gaussian quantum discord [58]
(17)$$D=z\left(\sqrt{\det A}\right)-z\left(\nu_{+}\right)-z\left(\nu_{-}\right)+z\left(\epsilon \right),$$
where $z\left(x\right)=\left(x+\frac{1}{2}\right)\ln\left(x+\frac{1}{2}\right)-\left(x-\frac{1}{2}\right)\ln\left(x-\frac{1}{2}\right)$, $\nu_{+}$ and $\nu_{-}$ are the symplectic eigenvalues which write as
(18)$$\nu_{\pm }=\sqrt{\frac{\Gamma \pm \sqrt{\Gamma^{2}-4\det\sigma_{AB}}}{2}},$$
where Γ=detA+detB+2detC and ϵ is defined by
(19)$$\epsilon =\frac{\sqrt{\det A}+2\sqrt{\det A\det B}+2\det C}{1+2\sqrt{\det B}}.$$If *D* is greater than 1, two modes’ quantum states cannot be separated. Additionally, the two modes may be in a separable state or an entangled state if the condition 0≤D<1 is met.

## 5. Results and Discusion

In this section, we show the results and discuss the evolution of quantum correlations of the system by considering experimentally accessible parameters reported in [62]: $\omega_{a}/2\pi =10$ GHz, $\omega_{m}/2\pi =10$ MHz, $\gamma_{m}/2\pi =100$ Hz, $\gamma_{a}/2\pi =\gamma_{b}/2\pi =1$ MHz, $g_{a}/2\pi =G_{b}/2\pi =3.2$ MHz, and at low temperature T=10 mK. $G_{b}=2\pi \times 3.2$ MHz implies the drive magnetic field $B_{0}\approx 3.9\times 10^{-5}$ T for $g_{b}\approx 2\pi \times 0.2$ Hz, corresponding to the drive power P=8.9 mW.

We present in Figure 2, the entanglement in the steady state of the three bipartitions, $E_{ab}$ (between the cavity and magnon mode), $E_{bm}$ (between the magnon and mechanical mode) and $E_{am}$ (between the cavity and mechanical mode) versus the detunings $\Delta_{a}$ and ${\tilde{\Delta }}_{b}$ in the presence of coherent feedback loop with the magnon self-Kerr nonlinearity. We observe, that the entanglement is very strong ($E_{ab}>1.3$, $E_{bm}>0.8$ and $E_{am}>1.3$) in comparison with the results in Ref. [62]. The maximum value of entanglement of the three bipartitions is improves via coherent feedback loop and the magnon self-Kerr nonlinearity when β=π. We remark, when $\Delta_{a}=-\omega_{m}$ and ${\tilde{\Delta }}_{b}=0.9\omega_{m}$ the entanglement $E_{ab}$ and $E_{am}$ are maximum while $E_{bm}\approx 0.2$.

In Figure 3 we plot there bipartite entanglements $E_{ab}$, $E_{bm}$ and $E_{am}$ as a function of the reflectivity τ and β. We remark that the entanglement is increasing with τ and β and it robust when β=π. Moreover, the entanglement is achieved its maximum value when $\gamma_{fb}=\gamma_{a}\left(1+2\tau \right)$.

In Figure 4 we plot the entanglements of the three bipartitions $E_{ab}$, $E_{am}$ and $E_{bm}$ versus different parameters. We remark that the existing of genuine tripartite entanglement when all bipartite entanglement are non-vanishing as illustrated in Figure 4. We notice, the entanglement is robust against temperature as depicted in Figure 4a and survive above 3 K. We observe that the entanglement of all the subsystem is diminishes due to decoherence phenomenon [68]. Moreover, the entanglement between photon-magnon and photon-phonon persists for temperature T>3 K and T≈2.5 K respectively, whereas, the entanglement between magnon-phonon vanishes at lower temperatures (T≈0.2 K) even this temperature is the maximum achieved in the Ref. [62]. One can say that the entanglement between photon-magnon and photon-phonon is stronger than the entanglement between magnon-phonon. The entanglement between photon-magnon and magnon-phonon increases with increasing the magnon self-Kerr nonlinearity coefficient ξ, instead the entanglement between photon-phonon decreases as illustrated in Figure 4b. The entanglement $E_{ab}\approx 0.25$ for $\xi =10^{7}$ Hz in comprising $E_{ab}\approx 0.125$ in comparison with the results in Ref. [62]. We remark in Figure 4c the enhancement of all three bipartitions entanglement by coherent feedback technique. The maximum value reached by entanglement between photon-magnon and photon-phonon is more significant than the one obtained in Ref. [62].

In Figure 5, we plot for each bipartite the entanglement, the Gaussian steering $S^{A\to B}$, $S^{B\to A}$ and the asymmetric steering versus the temperature *T*. The entanglement and steerability diminish quickly with temperature due to the decoherence phenomenon. We note, the one way quantum steering is more robust than two way quantum steering and it survive for a larger value of temperature *T*. The entangled state is not always steerable state instead steerable state must be entangled i.e., when $S^{A\to B}=S^{B\to A}>0$ and $E_{N}>0$ is the witnesses of existence of Gaussian two-way steering, such that the subsystem of two subsystem are entangled but are steerable only from *A* to *B* and from *B* to *A* [27] and no-way steering appears when $S^{A\to B}=S^{B\to A}=0$ and $E_{N}>0$ as depicted in Figure 5c. The measurement of Gaussian steering is always bounded by the entanglement $E_{N}$ as also discussed in [69]. Finally, the asymmetric steering SA is always less than ln(2), which is maximal when the state is nonsteerable in one-way i.e., $S^{A\to B}>0$ and $S^{B\to A}=0$ or $S^{A\to B}=0$ and $S^{B\to A}>0$ and it decreases with increasing steerability in either way [27]. In Figure 5a the steering from the photon mode to the magnon mode $S^{a\to b}$ has a similar behavior to $E_{N}$ it decreases from its maximum value to zero when T>3 K. Besides, one-way steering appears when T>0.2 K, i.e., $S^{a\to b}>0$ and $S^{b\to a}=0$ as expected in Figure 5a. Moreover, the steering from the magnon mode to the photon mode $S^{b\to a}$ is diminishes quickly to remains zero for T>0.2 K as depicted in Figure 5a. Otherwise, when the temperature T<0.2 K, the two-way steering occurs between optical mode and the magnon mode, i.e., $S^{a\to b}>0$ and $S^{b\to a}>0$ (S(ab)=0). The steerability between photon mode and the phonon mode is always remains one-way steering, i.e., $S^{a\to m}>0$ ($S^{m\to a}=0$) when T>0.2 K as implemented in Figure 5b. The steerability between the magnon mode and phonon mode approximately remains two-way and $S^{b\to m}>S^{m\to b}$ when T<0.10 K and no-way steering ($S^{b\to m}=0$ and $S^{m\to b}=0$ (S(bm)=0) when T>0.10 K as shown in Figure 5c.

Figure 6 shows the steady state entanglement and Gaussian quantum discord between the two modes with respect to temperature *T*. We remark that when *T* increases, the entanglement and Gaussian quantum discord between the two modes indeed degrade. This degradation can be attributed to the increased thermal noise and the resulting decoherence effects. Interestingly, even when the entanglement between the modes vanishes, the Gaussian quantum discord can still remain non-zero. Quantum discord quantifies the nonclassical correlations beyond entanglement and can persist in systems with vanishing entanglement. The disappearance of entanglement and the persistence of non-zero quantum discord reflect the fragility of entanglement under the influence of environmental degradation. Entanglement is a delicate quantum resource that requires careful control and isolation from the effects of the environment to maintain its coherence.

In Figure 7, we show the dynamics evolution of the entanglement and quantum discord between the two modes. We note that in a region, the negativity logarithmic is vanishing ($E_{N}=0$ for separable state) as expected in Figure 7, in contrary the Gaussian quantum discord *D* is non zero. This means that the Gaussian quantum discord is an important indicator about the quantum correlations. Besides, *D* is less than one (D<1) when $E_{N}=0$. Stationary entanglement between the two modes is achieved when $E_{N}$ remains constant with time.

## 6. Conclusions

In conclusion we have studied how coherent feedback loop improves the quantum correlations between three bipartite subsystems in the presence of the magnon self-Kerr nonlinearity in cavity magnomechanics systems. We quantify steerability by using Gaussian quantum steering and reveals that the Gaussian steering is limited by entanglement, indicating that the modes that can be steered are strictly entangled. However, it is important to note that the entangled modes are not necessarily steerable, meaning that the presence of entanglement does not guarantee the ability to control or steer the system. This implies that the relationships between steerability and entanglement in the considered system are closely interconnected. The entanglement serves as a resource that enables steering, but the ability to steer depends on additional factors beyond entanglement alone.We have found that there is a one-way steering between photon-magnon and photon-phonon, but the steerability between magnon-phonon is always two-way. Additionally, we have also observed that entanglement and steerability are robust against the temperature effects, with entanglement persisting above 3 K for photon-magnon, and approximately 2.5 K for photon-phonon. This suggests that these quantum properties can be maintained even at relatively high temperatures. Furthermore, we have studied the entanglement and Gaussian quantum discord in both steady and dynamical states and have shown that Gaussian quantum discord goes beyond entanglement, demonstrating the presence of quantum correlations that are not solely attributable to entanglement. However, we have also noted that, entanglement, steerability and Gaussian quantum discord, are fragile to thermal effects, implying that they are sensitive to changes in temperature. Our proposed scheme to enhance entanglement, which opens up possibilities for various applications in quantum information processing. This suggests that our work has potential implications for improving the performance of quantum communication and computation systems.

## Figures and Tables

**Figure 1 entropy-25-01462-f001:**
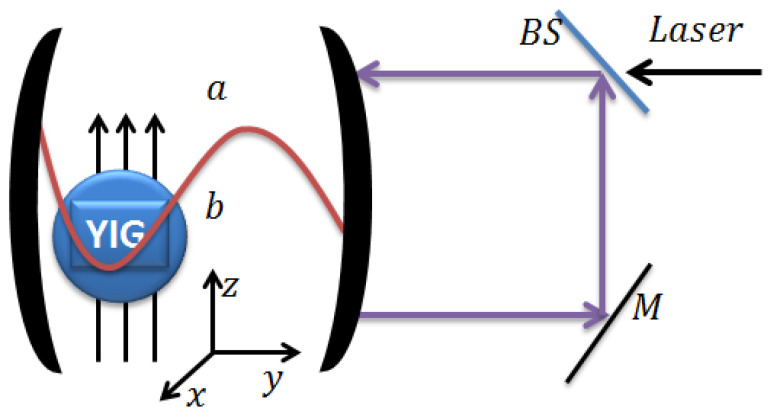
Schematic diagram of a single-mode cavity with a feedback loop and a YIG sphere with magnon self-Kerr nonlinearity. In order to improve the magnomechanical coupling, the magnon mode is directly driven by a microwave source, which is how the magnons are represented by the collective motion of numerous spins in a macroscopic ferrimagnet. The cavity is also driven by an electromagnetic field through an asymmetric beam splitter (BS) with amplitude ψE. We denote the transmission and reflection coefficients by ψ and τ respectively, and the phase shift generated by the reflectivity of the output field on the mirrors by β [44,59]. The photons and magnons of the cavity are coupled by dipole magnetic interaction, and the magnons and phonons are coupled by magnetostrictive interaction. A microwave field (not shown) is implemented to improve magnon-phonon coupling. The magnetic field of the cavity mode (along the x-axis), the driving magnetic field (in the y-direction), and the bias magnetic field (in the z-direction) are all common perpendiculars at the sphere YIG. Through an asymmetric beam splitter (BS), a laser light field entering the cavity is split asymmetrically. The output field is completely reflected on the *M* mirror, and the beam splitter sends some of the output field into the cavity.

**Figure 2 entropy-25-01462-f002:**
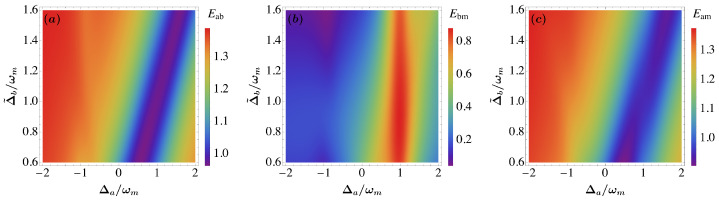
(**a**) Density plot of bipartite entanglement between photon and magnon modes $E_{ab}$, (**b**) magnon and phonon modes $E_{bm}$ and (**c**) cavity and phonon modes $E_{am}$ as function of normalized detunings $\Delta_{a}/\omega_{m}$ and ${\tilde{\Delta }}_{b}/\omega_{m}$ for τ=0.9, $G_{b}/2\pi =3.2$ MHz, T=10 mK, β=π and $\xi =\gamma_{a}$. See text for the other parameters.

**Figure 3 entropy-25-01462-f003:**
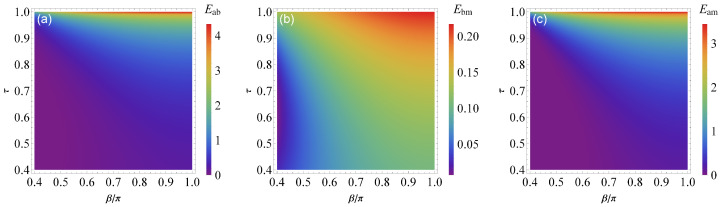
(**a**) Density plot of bipartite entanglement between photon and magnon modes $E_{ab}$, (**b**) magnon and phonon modes $E_{bm}$, and (**c**) cavity and phonon modes $E_{am}$ versus the reflectivity parameter τ and phase β for ${\tilde{\Delta }}_{b}=0.9\omega_{m}$, $G_{b}/2\pi =3.2$ MHz, $\Delta_{a}=-\omega_{m}$, and $\xi =\gamma_{a}$. See text for the other parameters.

**Figure 4 entropy-25-01462-f004:**
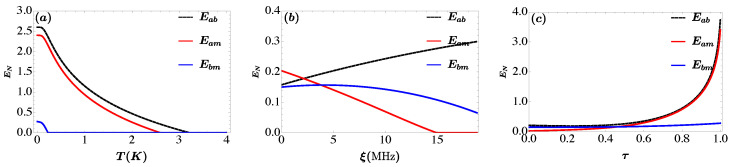
(**a**) Plot of photon and magnon modes ($E_{ab}$), cavity and phonon modes ($E_{am}$) and magnon and phonon modes ($E_{bm}$) as a function of temperature *T* (see the (**a**)), self-Kerr coefficient ξ (see the (**b**)) and reflectivity parameter τ (see the (**c**)). We take $G_{b}/2\pi =4.8$ MHz, $\Delta_{a}=-\omega_{m}$ and ${\tilde{\Delta }}_{b}=0.9\omega_{m}$. The reflectivity parameter is τ=0.98 and τ=0.4 in (**a**) and (**b**) respectively. In (**b**,**c**) the temperature is T=10 mK and in (**a**–**c**) the magnon self-Kerr nonlinearity coefficient is $\xi =\gamma_{a}$. See text for the details of the other parameters.

**Figure 5 entropy-25-01462-f005:**
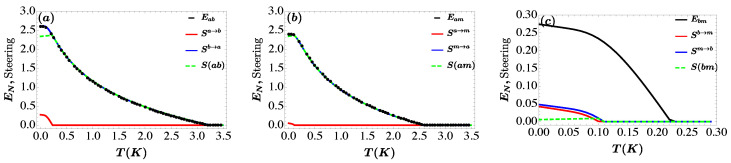
Plot of bipartite entanglement, Gaussian quantum steering and asymmetric quantum steering between (**a**) photon and magnon modes $E_{ab}$, $S^{a\to b}$ and $S^{b\to a}$ and S(ab), (**b**) cavity and phonon modes $E_{am}$, $S^{a\to m}$, $S^{m\to a}$ and S(am), and (**c**) magnon and mechanical modes $E_{bm}$, $S^{b\to m}$, $S^{m\to b}$ and S(bm), as a function of the temperature *T*. The parameters are $G_{b}/2\pi =4.8$ MHz, β=π, τ=0.98, $\xi =\gamma_{a}$, ${\tilde{\Delta }}_{b}=0.9\omega_{m}$ and $\Delta_{a}=-\omega_{m}$. See text for the details of the other parameters.

**Figure 6 entropy-25-01462-f006:**
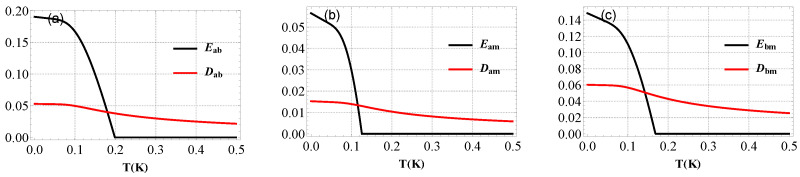
(**a**) Plot of bipartite entanglement $E_{ab}$ and Gaussian quantum discord $D_{ab}$ between photon and magnon modes (**b**) $E_{am}$ and $D_{am}$ between cavity and phonon modes and (**c**) $E_{bm}$ and $D_{bm}$ between magnon and mechanical modes as a function of the temperature *T*. The parameters are $G_{b}/2\pi =4.8$ MHz, β=π, τ=0.2, $\xi =\gamma_{a}$, ${\tilde{\Delta }}_{b}=0.9\omega_{m}$ and $\Delta_{a}=-\omega_{m}$. See text for the details of the other parameters.

**Figure 7 entropy-25-01462-f007:**
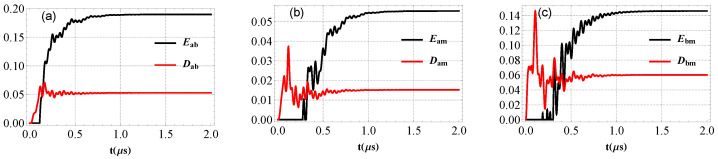
(**a**) Time evolution of bipartite entanglement $E_{ab}$ and Gaussian quantum discord $D_{ab}$ between photon and magnon modes, (**b**) $E_{am}$ and $D_{am}$ between cavity and phonon modes and (**c**) $E_{bm}$ and $D_{bm}$ between magnon and mechanical modes. The parameters are $G_{b}/2\pi =4.8$ MHz, T=10 mK, β=π, τ=0.2, $\xi =\gamma_{a}$, ${\tilde{\Delta }}_{b}=0.9\omega_{m}$ and $\Delta_{a}=-\omega_{m}$. See text for the details of the other parameters.

## Data Availability

The datasets used and/or analyzed during the current study are available from the corresponding author on reasonable request.

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
