# Peer review of "Feedback Control of Quantum Correlations in a Cavity Magnomechanical System with Magnon Squeezing"

_entropy, 2023, doi:10.3390/e25101462_

Round 1
Reviewer 1 Report
Please see the attachment

Please see the attachment
Author Response
Dear Reviewer,
Thank you very much for the time and efforts you took to process our manuscript in Entropy, entitled "Coherent feedback control of quantum correlations in cavity magnomechanical system with magnon squeezing". We appreciate the insightful comments provided and have taken them into careful consideration as reflected in our revised version.
Our point-to-point replies are enclosed below along with a list of changes. We think that our revised manuscript has addressed all the comments raised by the referee and hope that it is now suitable for publication in Entropy.
Yours faithfully,
Authors

Reviewer 2 Report
In the manuscript. The authors investigate the quantum correlations in cavity opto-magnomechanical system by using the coherent feedback loop in the presence of magnon kerr nonlinearity. They show the entanglement and steering affected by coherent feedback and kerr nonlinearity. It seems that the manuscript publishable in Entropy. Before publication, I suggest the authors to add following aspects.
1.I found the stability condition is missing. Since the kerr nonlinearity is linearized into a degenerate-parametric form (squeezing term), the system is unstable. The parameters should satisfy the stable condition. I suggest that the parameters of their figures are not in the stable region. Authors should add discussion about the stable of the system and make assure all the plots in stable region.
2.kerr nonlinearity may be arose from the magnetocrystalline anisotropy of YIG spheres. The corresponding statement should be added.
English is ok.
Author Response
Dear Reviewer,
Thank you very much for the time and efforts you took to process our manuscript in Entropy, entitled "Coherent feedback control of quantum correlations in cavity magnomechanical system with magnon squeezing". We have carefully studied the reports and are pleased to see that generally positive about our work. We appreciate the insightful comments provided and have taken them into careful consideration as reflected in our revised version.
Our point-to-point replies are enclosed below along with a list of changes. We think that our revised manuscript has addressed all the comments raised by the referee and hope that it is now suitable for publication in Entropy.
Yours faithfully,
The Authors

Reviewer 3 Report
Please see the attached PDF file named "Report entropy-2379763 R1.PDF"

Please see the attached PDF file named "Report entropy-2379763 R1.PDF"
Author Response

(The authors gave the same response as above.)

Reviewer 4 Report
I also attach this review as a pdf.
In the manuscript, the authors study a three-mode system including a microwave
cavity, a magnon mode and a mechanical oscillator, coupled pairwise. The paper
largely follows [PRL 121, 203601 (2018)] (cited as Ref.[53]) with addition of
self-Kerr nonlinearity to the magnon mode, and a coherent feedback.
I have read thoroughly the present manuscript which happens to be a substantial
piece of work. Nevertheless, the overall impression is rather negative for a
number of reasons.
1. In my opinion, the results are not discussed properly. The present
manuscript aims at generalization of the results of [53] by adding self-Kerr
nonlinearity and feedback, and the authors do observe increase of the
figures of merit from the additions. There is, however, very limited
discussion regarding how exactly these additions contribute positively. In
particular, what is the role of the feedback: does it implement an effective
cooling or something else? Similarly, what is the role of the nonlinearity,
is it to perform effective squeezing? Are both additions necessary
(nonlinearity and feedback)?
2. I do not find the way the feedback is introduced, satisfactory. It is
implied that there are additional mirrors that reroute the leaking field
back to the input, however, I don’t think it is explained explicitly in the
text. Typically, the coherent feedback field is assumed to be separate from
the primary cavity input (e.g., orthogonally polarized; see e.g. [42]).
Could the authors justify properly the model they use?
3. there are errors in the manuscript, in particular, in the definitions of
logarithmic negativity and steering:
- Λ⁻ is a symplectic eigenvalue of the covariance matrix of a partially
transposed state which is not exactly what the authors write
- the definition of ? is wrong, should be X = det A + det B − 2det C, the
authors missed 2 before det C; note that on p.7 introducing the symplectic
eigenvalues of the original state (no partial transposition), analogous
formula has factor 2
- the systems are entangled when Λ⁻ < 1/2, which is the opposite of what the
authors write
- the definition of steering (Eq.(15)) is notably different (has a spurious
factor of 4 in the denominator) from Eq.(5) of Ref.[34]
These would significantly impact the validity of the results, however, are very
likely to be just typos, given the large amount of ordinary typos and
inaccuracies in the manuscript.
4. there is indeed a very large amount of grammar mistakes and typos, see a
list below. In particular, the authors should check capitalization of names
in the bibliography: I saw Einstein, Podolsky, Rosen, Bell, and Raman
starting from a lowercase letters. Something is wrong with Refs. [55,56].
5. finally, there were works studying steering in pulsed optomechanics that the
authors can find interesting
[1] S. Kiesewetter, Q. Y. He, P. D. Drummond, and M. D. Reid, Scalable Quantum
Simulation of Pulsed Entanglement and Einstein-Podolsky-Rosen Steering in
Optomechanics, Phys. Rev. A 90, 043805 (2014).
[2] N. Vostrosablin, A. A. Rakhubovsky, U. B. Hoff, U. L. Andersen, and R.
Filip, Quantum Optomechanical Transducer with Ultrashort Pulses, New J. Phys.
20, 083042 (2018).
Grammar, typos and other minor issues:
- ‘sensitiveness’ in the abstract seems inappropriate, likely has to be
‘sensitivity’
- the acronym ‘YIG’ is defined twice, in two subsequent sentences of the
second paragraph on p.1; then again in Sec. 2. Also, it is probably
unnecessary to write YIG in brackets throughout the full text.
- ‘very lower’ should be ‘much lower’
- ‘used to quantify how much the two entangled bipartite states are steerable’
is formally incorrect. Different quantum states can not be steerable,
different systems can be.
- the sentence ‘entanglement for two-mode continuous-variable (CV)’ on p.2,
line 52 is incomplete, perhaps the word ‘system’ is missing
- first line p.3: ‘is described by the form’ should be ‘has the form’
- the rotating frame introduced on p.3, line 77 should perhaps include the
optical drive frequency as well
- it is not clear why square root of gammas is typeset differently in
equations for a and b in Eqs.(2)
- the quantities ψ, τ are introduced in an ambiguous manner
- ‘assosiatetd’ → associated (p.3, l.96)
- ‘corresponding’ → corresponding (l.98)
- ‘analytical’ → analytically (p.4, l.107)
- ‘normalize’ → normalized (Fig. 2 caption)
- there is no imaginary unit near ψ in the first of Eqs. (2), but it is there
in Eq. (8)
- ‘steady state evolution’ sounds incorrect, there’s apparently not much
evolution in the steady state
- the full first sentence of Sec. 4 is grammatically incorrect
- the sentence ‘Characterizing, quantifying .. are one’ is grammatically
incorrect, should be ‘is one’. Also, I don’t think it is appropriate to call
these problems ‘problematic issues’
- the font size of insets in most figurs is too low
- on p.7 ‘steady state of … entanglement’ is incorrect, should be
‘entanglement in the steady state’
- ‘in comprising’ → ‘in comparison’ (on pp.7 and 8)
- in line 137 the sentence is incomplete (ends with ‘as shown’)
- ‘genuig’ → ‘genuine’
- ‘is vanishes’ → ‘vanishes’
- ‘is very important than’ is grammatically incorrect
- ‘is approximately remains’ is incorrect
- p.8, line 184 ‘T enhances’ → ‘T increases’, ‘can be explain’ → ‘can be
explained’

Author Response

(The authors gave the same response as above.)

Round 2
Reviewer 1 Report
No
Nothing
Author Response
We thank the Referee for his/her carefully review. We addressed all the suggestion in order to further improve the caliber of the manuscript.
Reviewer 3 Report
Please see the attached pdf file.
The English language is approriately written, but it can be improved as indicated in the report.
Author Response
We thank the reviewer for his/her insightful and highly relevant questions and comments. We Agree; clarification added both in the introduction and on the body of the manuscript.

Reviewer 4 Report
The criticisms that I presented for the original submission, mostly remain for the resubmission, as the authors have made only marginal corrections to the manuscript, which have introduced new technical errors.
To summarize, my two main concerns were in that
1. The roles of neither feedback nor non-linearity were not discussed properly in the manuscript. The question of whether these both are required for the results is not answered.
2. The model of the feedback was not justified. The authors' model does not use all of the degrees of freedom available in an experiment and hence this model is likely suboptimal.
These two concerns were not addressed in the resubmission, so I do not think the manuscript has been sufficiently improved to be published.
In fact, the authors write in the resubmission cover letter that they '... have added a detail analysis on on how the feedback mechanism influences the system dynamics... ', '...have revised the manuscript to provide a more explicit and detailed description of how the feedback mechanism is implemented...' but I could not find these revisions in the resubmission. Furthermore, the authors write in the cover letter that they have fixed capitalization in References (they have not, Einstein, Podolsky, Rosen, Laguerre, Horne, Shimony, and Holt are not capitalized), and that they have added the two references I suggested, but those are not there in the resubmission. This contradiction makes me assume that the authors have mistakingly submitted not the final version of the corrected manuscript.
Moreover, the minor corrections the authors made, have introduced a lot of mistakes
1. The authors attempted to change the notation for the mechanical quadratures from $x,y$ to $q,p$, but the old notation ($x,y$) is yet present in Eqs. (1,2) and an inline eqn on p. 5. This time, the old quadratures are undefined but bear a resemblance to the axes labels in Fig.1.
2. I suggested correcting the (incorrect) definition of $\Lambda^-$. The authors have kept the incorrect one, but have added a new, correct one a few lines below.
3. Minor typos:
a. p.2, l.71: 'has takes the form' should be either has or takes
b. p.7, l.145: 'three bipartite' should be 'three bipartitions'
There are suspicious cases of pluralization that do not sound natural, like entanglements, also see the main review.
Author Response
We would like to thank Referee for his/her interesting remarks and suggestions and appreciate the reviewer for pointing out these typos and we have thoroughly checked our manuscript and tried to correct all the typos and amendments throughout the manuscript.

Round 3
Reviewer 4 Report
I do not find the minor corrections the authors have introduced, sufficient. The changes are very nominal and incremental while the paper, in my opinion, requires a major revision, as I indicated in the two previous rounds. It does not seem that the authors intend to address the two main criticisms I had (justify the used feedback model and discuss the role of the nonlinearity and feedback) in anything but generic words, so I don't think further review makes sense. I have to admit that the introduction of the feedback is now described in a much more consistent way.
In addition, the definition of Lambda after Eq.(14) 'the smallest symplectic eigenvalue of partial transposed covariance matrix' is incorrect both grammatically and mathematically (it is apparently impossible to do a partial transposition of a 2d-array) and was so in the original submission which I pointed out.
The pluralization 'steerabilities' sounds very strange to me.
Author Response
Dear Referee,
Thank you for the report for our manuscript “entropy-2379763”. We appreciate his/her critical reading and feedback on our work. We have amended our manuscript adding a discussion on the point raised by the Referee.
We modified a paragraph in the Introduction on second page of the first paragraph about some generic argument about the use of a feedback model in the cavity magnomechanical system with magnon squeezing and nonlinearity.
We have also amended the definition of Lambda after Eq.(14) and fixed the both grammatically and mathematically.
We agree with the Referee that “The pluralization 'steerabilities' sounds very strange to me.”
We fixed to steerability.
